# Behind the Scenes of COVID-19 Vaccine Hesitancy: Psychological Predictors in an Italian Community Sample

**DOI:** 10.3390/vaccines10071158

**Published:** 2022-07-21

**Authors:** Sofia Tagini, Agostino Brugnera, Roberta Ferrucci, Alberto Priori, Angelo Compare, Laura Parolin, Gabriella Pravettoni, Vincenzo Silani, Barbara Poletti

**Affiliations:** 1Department of Neurology and Laboratory of Neuroscience, Istituto Auxologico Italiano, IRCCS, 20149 Milan, Italy; vincenzo@silani.com (V.S.); b.poletti@auxologico.it (B.P.); 2Department of Human and Social Sciences, University of Bergamo, 24129 Bergamo, Italy; agostino.brugnera@unibg.it (A.B.); angelo.compare@unibg.it (A.C.); 3Aldo Ravelli Center for Neurotechnology and Experimental Brain Therapeutics, Department of Health Sciences, International Medical School, University of Milan, 20122 Milan, Italy; roberta.ferrucci@unimi.it (R.F.); alberto.priori@unimi.it (A.P.); 4Neurology Unit I, ASST Santi Paolo e Carlo, 20142 Milan, Italy; 5Department of Psychology, University Milano Bicocca, 20126 Milan, Italy; laura.parolin@unimib.it; 6Department of Oncology and Hemato-Oncology, University of Milan, 20122 Milan, Italy; gabriella.pravettoni@ieo.it; 7Applied Research Division for Cognitive and Psychological Science, IEO, European Institute of Oncology, IRCCS, 20141 Milan, Italy; 8Dino Ferrari Center, Department of Pathophysiology and Transplantation, University of Milan, 20122 Milan, Italy

**Keywords:** COVID-19, vaccine hesitancy, psychological antecedents, psychological predictors

## Abstract

Psychological variables may be crucial in favoring or discouraging health-related behaviors, including vaccine acceptance. This study aimed to extend the previous literature by outlining the psychological profile associated with COVID-19 vaccine hesitancy in a sample of Italian citizens. Between April and May 2021, 1122 Italian volunteers completed a web survey on COVID-19 vaccine acceptance, also including several self-reported psychological measures. A multiple hierarchical logistic regression analysis was performed to identify the psychological variables associated with vaccine hesitancy. Low confidence in COVID-19 vaccine efficacy and safety, low collective responsibility, high complacency, and high calculation (i.e., extensive information searching, and costs–benefit estimates) predicted higher hesitancy. Our results suggest that to be effective, vaccine-related communications should be as clear, understandable, and sound as possible, preventing the spreading of misunderstandings, or even fake information, that may foster people’s insecurities and distrust. Furthermore, the advantages and necessity of vaccination, both at the individual and community-level, should be clearly emphasized. Efficacious vaccine-related communications may be crucial, not only to maintain an adequate immunity rate for COVID-19, but also to inform policymakers and public authorities in the case of possible future infectious outbreaks.

## 1. Introduction

The COVID-19 vaccination campaign was a crucial turning point in the pandemic’s containment [1,2]. By the end of February 2022, 89.27% of those people suitable to receive COVID-19 vaccination had completed the vaccination course in Italy [3]; in the same period, 61.5% of eligible European citizens were fully vaccinated, whereas in other Western Countries the acceptance rate ranged from 63.9% in the US to 94.4% in Australia [4]. Eventually, a satisfactory immunization rate was achieved in Italy; however, spontaneous adherence was not so obvious [5,6]. It is likely that the increasing governmental limitations for those not vaccinated (e.g., concerning leisure activities, but also access to public transport and workplaces) were critical in incrementing the overall vaccine uptake [7]. Nevertheless, why did many people procrastinate, and some refuse the COVID-19 vaccine, despite its proven efficacy and safety?

Vaccine hesitancy, which ranges from a delay in acceptance, to active refusal of vaccines [8], has been associated with several psychological constructs. Focusing on our country, at an early stage of the pandemic (before COVID-19 vaccines were available), low health engagement [9], low perceived risk for the disease, conspiracy beliefs, and overall mistrust of science, vaccines, and medical information [10,11] were associated with lower intention to be vaccinated against COVID-19. Likewise, the level of risk perception, trust in government and science, misinformation, and conspiracy beliefs were related to vaccine hesitancy after COVID-19 vaccines were released [12,13]. Furthermore, Italians’ willingness to receive COVID-19 vaccine, after its released, was positively related to confidence in vaccine efficacy and safety, higher collective responsibility, and lower information searching and cost–benefit estimates (i.e., a component known as calculation) [14]. Indeed, confidence, collective responsibility, and calculation were included in the five most relevant psychological antecedents of vaccination (in general), together with complacency and the perceived constraints of vaccination [15].

Worldwide, investigations [16,17,18] have generally mirrored what was observed in Italy and point to the possible additional role of the perceived locus of control and individuals’ personal values, such as social dominance and authoritarianism [19]. In addition, personality traits, including agreeableness and consciousness, were associated with COVID-19 vaccine acceptance, both before [19] and after [13] the vaccinations began, suggesting that more agreeable and conscious people are less likely to be hesitant. 

To sum up, several psychological constructs have been related to COVID-19 vaccine hesitancy but by independent studies; thus, possibly providing only a partial picture of the whole scenario and interplay between these variables. Furthermore, some of the previous investigations focused on specific populations (e.g., university students [13], healthcare workers [16]), limiting the generalizability of these results to the overall population. This study aimed to extend the previous literature by outlining more comprehensively the psychological profile associated with COVID-19 vaccine hesitancy in Italy, evaluating in a sample of Italian citizens those psychological variables previously, but independently, reported to be significant predictors of COVID-19 vaccine hesitancy. For this purpose, several self-report psychological measures were collected as part of a web survey on COVID-19 vaccine acceptance released in Italy between April and May 2021.

## 2. Materials and Methods

### 2.1. Participants

A priori sample size estimation performed with G*Power software [20] showed that 1045 participants would be sufficient to achieve 95% of statistical power, considering a logistic regression analysis with moderately correlated predictors (r^2^ = 0.25), an average odds ratio of 1.30, and a given alpha level of 0.05.

Inclusion criteria were being aged 18 or older and being Italian residents. A total of 1122 online records were obtained; 1079 participants were included in the final sample (mean age: 46.91 ± 15.59 years, age range: 18–89 years; females *n* = 788, 73.2%), after removing a few duplicated cases (*n* = 4), those who did not agree with the privacy regulation (*n* = 33), and those who did not live in Italy (*n* = 6). Most of the sample were living in Northern Italy (*n* = 999; 92.6%), attended at least high school (*n* = 410; 38%) or had a university degree (*n* = 680; 63%), was in a relationship (*n* = 810; 75.1%), and reported a medium-level socioeconomic status (*n* = 680; 63%). 

### 2.2. Procedure 

An online, cross-sectional survey on COVID-19 vaccine acceptance in Italy was conducted between April and May 2021, approximately three months after the start of the vaccinal campaign in our country. Questionnaires were created and delivered using Google Forms (©Google). Participation was anonymous. A snowball convenience sampling procedure was adopted. The survey was sponsored through institutional media, social networks, and authors’ personal and professional contacts. Before completing the questionnaire, participants gave their digital informed consent, declaring being older than 18 years and to have read and accepted the privacy regulation. The study was conducted in accordance with the ethical principles of the Declaration of Helsinki and was approved by the Ethical Committee of the IRCCS, Istituto Auxologico Italiano.

### 2.3. Measures

An online form was developed according to the previous literature on vaccine hesitancy, in general, and relative to COVID-19 vaccine [8,21,22,23,24]: those predictors identified as most influential were included and assessed through ad hoc questions or standardized measures. Eventually, various items were included in the form, including sociodemographic, health-related, behavioral, and psychological information (see Appendix A for details on the composition of the overall survey). Information and measures to be included in the present study were selected in line with the study aim and according to the available literature on the psychological predictors of COVID-19 vaccine hesitancy [10,11,13,14,16,17,18,19]; these are detailed below. Furthermore, essential sociodemographic information was included in the analysis: specifically, age, sex, place of living, civil status, level of education, profession, socioeconomic status, and certified COVID-19 contagion.

Acceptance of COVID-19 vaccine was investigated by asking participants whether (i) they had already received the shot, (ii) they would like to be vaccinated in the future, (iii) they were doubtful, or (iv) they will not get the shot for sure. In addition, we asked participants whether they had, or they would, get the COVID-19 vaccine only because forced by law (i.e., since April 2021 COVID-19 vaccination was a prerequisite for practicing healthcare professions in Italy). 

Risk perception for COVID-19 was investigated with a 5-item questionnaire previously developed by our research group, which proved to have good internal validity [20]. As illustrated in Appendix A, three items investigated the perceived harmfulness of COVID-19 to one’s health, that is the disease severity; and two items assess the perceived likelihood of getting sick, that is personal vulnerability. Items were rated on a 5-point Likert-type scale (1 = totally disagree; 5 = totally agree). In accordance with the previous literature [25,26,27], the overall risk perception for COVID-19 was computed as the product of severity and vulnerability. In this study, the Cronbach’s alpha was good (α: 0.74). 

Italian validated translations of standardized instruments, with verified psychometric properties, were adopted to measure additional psychological constructs. Those instruments that were not available in Italian were translated using a forward and backward translation procedure [28]; the Italian translations adopted are reported in the Appendix A. Moreover, confirmatory factor analyses were performed to probe the internal validity of the Italian version of the translated measures (see the Appendix A). For all measures, Cronbach’s alpha or inter-item correlation coefficients were provided as indices of internal reliability. Standardized instruments were scored according to guidelines. 

Confidence, complacency, constraints, calculation, and collective responsibility were measured with the 5C Psychological Antecedents of Vaccination scale (5C-PAV) [15]. The questionnaire includes 15 items rated on a 7-point Likert-type scale (1 = totally disagree; 7 = totally agree); the items wording was adapted to the COVID-19 vaccine. Higher scores indicate higher confidence, constraints, calculation, and collective responsibility, but low complacency. The original items were translated into Italian (see Appendix A) using a forward and backward translation procedure [28]; a confirmatory factor analysis (CFA) evidenced that the Italian version of the 5C-PAV scale had good internal validity (see Appendix A). In this study, the Cronbach’s alphas of the five subscales were fair to good (α range: 0.63–0.87).

Personality traits were measured with the Italian version of the Ten-Item Personality Inventory (TIPI) [29]. The questionnaire includes 10 items, rated on a 7-point Likert-type scale (1 = totally disagree; 7 = totally agree), which measure five personality traits according to the big five personality dimensions: extroversion, agreeableness, conscientiousness, emotional stability, and openness to experiences. In line with the previous literature [13,19], agreeableness and conscientiousness were included in the analysis as possible predictors of vaccine hesitancy. In this study, the inter-item correlation coefficients were good (agreeableness: 0.208; conscientiousness: 0.328); inter-item correlation coefficients in the range of 0.15–0.50 indicate the good internal consistency of a scale [30]. 

The health-related locus of control was measured with the Italian version of the Health Locus of Control scale (H-LoC) [31]. The questionnaire includes 13 items rated on a 5-point Likert-type scale (1 = totally disagree; 5 = totally agree), which measure the participants’ perception of having direct or indirect control over one’s health, namely an internal or external locus of control. The H-LoC scale is composed of three subscales: Internal LoC (control is attributed to oneself; 8 items), External LoC God (control is attributed to transcendental entities; 2 items), and External LoC Others (control is attributed to significant people; 3 items). In this study, the Cronbach’s alphas of the three subscales were fair to good (α range: 0.62–0.84).

Conspiracy beliefs were evaluated with the Conspiracy Mentality Questionnaire (CMQ) [32]. This is a 5-item self-report measure of generic conspiracy beliefs. Participants indicated on an 11-point scale how likely they thought each item was to be true, from 0 (0%–certainly not) to 10 (100%–certain); each point was labeled, with increasing probabilities in steps of 10 percentage points. The original items were translated into Italian (see Appendix A) using a forward and backward translation procedure [28]; a confirmatory factor analysis (CFA) evidenced that the Italian version of the CMQ had good internal validity (see Appendix A). In this study, the Cronbach’s alpha was good (α: 0.85).

Personal values were measured with The Portrait Values Questionnaire (PVQ)–European Social Survey version [33]. The PVQ includes 21 short verbal portraits of different imaginary people. Each portrait describes a person’s goals, aspirations, or wishes, which point implicitly to the importance of a specific human value (e.g., “Thinking up new ideas and being creative is important to him/her. He/she likes to do things in his own original way”). Participants say how much they recognize themselves in the described people using a 6-point Likert-type scale (i.e., 1 = not like me at all; 6 = very much like me). Respondents’ values are inferred from their self-reported similarity to the people described in terms of certain values. Overall, ten subscales can be computed, corresponding to ten different values: power, achievement, hedonisms, stimulation, self-direction, universalism, benevolence, tradition, conformity, and security. In line with the previous literature and the study aims, we considered self-direction (i.e., creativity, freedom, independence, curiosity), benevolence (which is related to the enhancement of the welfare of close people), conformity (i.e., meaning the restraint of actions, inclinations, and impulses likely to upset or harm others and/or to violate social expectations or norms), and security (i.e., seeking safety, harmony, and stability of society, relationships, and self). For each value, the score was computed as the mean of the ratings given to the items of the subscale, minus the mean scores across all the 21 items of the scale; this, allows the correction for possible individual differences in the use of the response scale (Schwartz, 2003). In this study, the inter-item correlation coefficients of most subscales were good (range: 0.22–0.39). The only exception was the subscale benevolence, whose inter-item correlation coefficient was above the cut-off (mean correlation = 0.52), but with a Cronbach’s alpha of 0.69.

### 2.4. Statistical Analysis

We identified two groups according to participants’ willingness to be vaccinated. Participants who had already had the shot or who were willing to be vaccinated represented the favorable group; those who were doubtful, who were certainly not going to be vaccinated in the future, or who had received (or would receive) the shot only because it was mandatory for working represented the hesitant group. Data were initially analyzed using descriptive statistics, including means, standard deviations, frequencies, and percentages, and preliminary screened for assumptions. 

In line with the study aims, a multiple hierarchical logistic regression analysis was performed to identify the psychological variables associated with vaccine hesitancy. Participants’ vaccine-related behavior was the dependent variable (favorable vs. hesitant). Since sociodemographic variables have been reported to be associated with vaccine hesitancy [34], in Block 1 of the regression model, we preliminary introduced age (in years), sex (clustered in female or male), civil status (i.e., single or in a relationship), education (clustered in pre-university, university, and post-university education), socioeconomic status (i.e., low, medium, high), being a healthcare professional (i.e., participants who reported to work as physicians, nurses, physiotherapists, occupational therapists, speech therapists, behavioral therapists, orthoptists, psychiatric rehabilitation technicians, biomedical technicians), and being infected with COVID-19 as possible sociodemographic predictors. In Block 2, risk perception for COVID-19, the 5C PAV subscales (i.e., 5C-PAV confidence, 5C-PAV complacency, 5C-PAV constraints, 5C-PAV calculation, and 5C-PAV collective responsibility subscales), TIPI agreeableness, TIPI conscientiousness, H-LoC internal, H-LoC external God, H -LoC external others, the CMQ total score, the PVQ–21 conformity, PVQ–21 benevolence, PVQ–21 self-direction, and PVQ–21 security subscales were entered as possible psychological predictors of vaccine hesitancy. This analytical approach allowed us to examine the predictive role of psychological dimensions over and above all other sociodemographic variables.

The assumption of multicollinearity was assessed by computing and examining both the variance inflation factor (VIF) and tolerance values. The presence of strong multicollinearity is suggested by values above 10 and below 0.1, respectively [35]. Statistical analyses were performed using SPSS statistical software (version 20.0; SPSS Inc., Chicago, IL, USA). All tests were two-sided and a *p*-value ≤ 0.05 was considered significant; effect sizes were interpreted according to guidelines [36].

## 3. Results

### 3.1. Preliminary Analysis

We initially screened data for assumptions. As for the presence of univariate outliers, we found few cases in the H-LoC internal, and in the complacency, constraints, and collective responsibility subscales of the 5C-PAV. All outliers were brought into range [37]. According to their skewness and kurtosis values, all 5C-PAV subscales were non-normally distributed, and the violation of the assumption was corrected through square-root, log10, or reflect and inverse transformations [37]. Finally, we found a total of 15 multivariate outliers, which were removed from subsequent analyses [37]. In our sample, a total of 136 individuals (12.8%) were hesitant towards the COVID-19 vaccine, while the remaining 928 (87.2%) were favorable towards it. Descriptives for all psychological variables, for the total sample as well as separately for each group, are reported in Table 1.

### 3.2. Main Analysis

We tested for the significant sociodemographic and psychological predictors associated with vaccine hesitancy through a multiple hierarchical logistic regression. The full model containing all predictors was statistically significant, χ^2^ (25) = 430.83, *p* < 0.001, explaining between 33.4% (Cox and Snell R^2^) and 62.8% (Nagelkerke R^2^) of the variance in vaccine hesitancy and correctly classifying 92.5% of cases. 

As shown in Table 2, considering the sociodemographic predictors associated with vaccine hesitancy, participants who were in a relationship were 0.41 times less likely to be favorable towards the vaccine than those alone; further, those who reported a medium socioeconomic status were two and a half times more likely to be favorable than those with a low socioeconomic status. Finally, those who had been infected with COVID-19 were 0.32 times less likely to be favorable. Among the psychological predictors, only the confidence, complacency, calculation, and collective responsibility 5C-PAV subscales made a unique and statistically significant contribution to the model. That is, for every unit increase in the predictor, those who believed that the COVID-19 vaccine is effective and safe (i.e., high confidence) and those who were more willing to protect others through herd immunity (i.e., high collective responsibility) were 47.62 and 18.45 times more likely to be favorable, while those who reported a lower perceived need and usefulness of the COVID-19 vaccine (i.e., low complacency), made more extensive information searching and costs-benefits estimates on COVID-19 vaccine (i.e., high calculation) were 0.03 and 0.26 and times less likely to be favorable towards the COVID-19 vaccine, respectively. These results were significant while controlling for all other variables in the model. A sensitivity analysis, removing from the hesitant group those who were forced to get a vaccine (*n* = 34), evidenced similar results for the psychological variables. 

## 4. Discussion

The study aimed to extend the previous literature concerning the possible psychological antecedents of vaccine hesitancy, investigating the psychological predictors of COVID-19 vaccine refusal or procrastination in a sample of Italian citizens between April and May 2021. Most of the sample (87%) had already been vaccinated, or were certainly intending to. The observed acceptance rate for COVID-19 vaccine is in line with previous studies on the Italian population [14,38]; however, it seems particularly high if compared with the records from other Western Countries [23,39]. The different timing of investigation relative to COVID-19 vaccine availability may explain this observation: a lot of the literature available worldwide on the topic refers to participants’ acceptance of a hypothetical vaccine against COVID-19. However, people’s attitudes and beliefs may have changed in face of the real chance of receiving a COVID-19 vaccine. For instance, the reported collateral effects likely influenced the acceptance rate once the vaccine was released. In addition, individuals’ willingness to receive the vaccine may have depended on the pandemic fluctuation and the relative perceived risk of infection; indeed, Caserotti and colleagues (2021) reported that people’s hesitancy increased during the national lockdown compared to the pre-lockdown and the re-opening phase in Italy.

Concerning the possible predictors of COVID-19 vaccine hesitancy, few sociodemographic variables were found to significantly affect participants’ vaccine acceptance in the regression model adopted. Participants in a relationship were slightly less likely to be favorable than those who were not. We speculate that individuals who are not in a relationship possibly seek more social interactions and others’ company outside the family. Thus, they may be more inclined to get vaccinated since they perceive the COVID-19 vaccine as necessary to restore usual networking activities, especially since COVID-19 containment measures in Italy significantly penalized gatherings with non-family members. Furthermore, people who reported a medium-level socioeconomic status were more likely to get vaccinated than those reporting a low-level socioeconomic status. A lower socioeconomic status has been previously related to higher hesitancy for the COVID-19 vaccine [34]; we speculate that higher difficulties in understanding vaccine-related communications and, possibly, higher distrust in the health system and/or policymakers [40] may be related to vaccine hesitancy in those with the lowest socioeconomic status. Nevertheless, the relationship between socioeconomic status and vaccine hesitancy (in general) has been reported to be controversial [41]; future studies might clarify this issue and the possible reasons behind the association between vaccine hesitancy and socioeconomic status. Moreover, participants who already got COVID-19 were less prone to be vaccinated, likely because a low perceived vulnerability to a re-infection. 

Focusing on the psychological variables of interest, we found that four of the five antecedents theoretically related to vaccination [15] significantly predicted individuals’ COVID-19 vaccine-related behavior in our sample. Not surprisingly, the strongest predictor of people’s choice to get vaccinated against COVID-19 was the perceived efficacy and safety of the COVID-19 vaccine. The more people believed that the COVID-19 vaccine was safe and efficacious and that public authorities acted in the best interest of the community, the greater the likelihood of being favorable. Therefore, policymakers should ensure the clearest and the soundest vaccine-related communications, which should be easily understandable throughout the entire population, preventing the spreading of incoherent, and even fake, information that may foster people’s insecurities. 

In addition, perceived collective responsibility was a strong positive predictor of COVID-19 vaccine acceptance, suggesting that the more people were willing to protect others through herd immunity and perceived COVID-19 vaccination as a collective duty, the more they were in favor of COVID-19 vaccine. The practical implications of this finding in promoting vaccine acceptance may be twofold. On one hand, strengthening the idea of vaccination as an act of social responsibility (i.e., a social norm to comply with to maintain the community approval) may reduce hesitancy in those slightly hesitant. However, this might be effective only to a certain extent; if the more hesitant people are also those lacking altruism, they might not be sensitive to these messages. Thus, emphasizing the personal benefits related to COVID-19 vaccination may be more efficacious. Furthermore, we observed that those who thought that the COVID-19 vaccine was unnecessary because getting COVID-19 was not easy or dangerous were slightly more likely to be hesitant. Accordingly, vaccine-related communications should specifically address and clarify the advantages and necessity of vaccination, both at the individual and community-level. Finally, those who engaged more in cost–benefit estimations and information seeking were slightly more likely to refuse or procrastinate about the COVID-19 vaccine. This might be surprising; however, Betsch and colleagues [15] had already noted that the decision-making behind calculations may not be skillful, potentially leading to biased decisions (i.e., high vaccination risks, low disease risks). The longer the research of information, the higher the probability of encountering questionable sources of information, increasing individuals’ doubts and insecurities. Moreover, meticulous calculations may take time, delaying vaccine uptake. Clear, sound (e.g., coherent, and not contradictory), concise, and honest vaccine-related communications may favor solid knowledge and quick, efficient decisions; also, this manner of communicating may favor individuals’ trust in science, which has been recently reported to be a crucial mediator of COVID-19 vaccine acceptance [42]. 

These findings are in line with the preliminary evidence available on the psychological predictors of the COVID-19 vaccine [14,16,17,18]. However, we may note that none of the other psychological variables considered in the present work significantly predicted COVID-19 vaccine hesitancy. Specifically, individuals’ values of conformity, benevolence, and security did not significantly predict vaccine hesitancy, although they may conceptually resemble the collective responsibility and confidence antecedents of vaccination, respectively. Nevertheless, COVID-19 vaccination represents a unique and unusual circumstance; hence, people’s general attitudes and overall values in typical situations may differ, depending on individual beliefs and feelings concerning the COVID-19 vaccine. Furthermore, this study focused on the Italian population, but vaccine-related behaviors and their determinants might differ across countries and cultures [43], possibly explaining the previous different results [19]. Furthermore, antecedents of vaccine hesitancy may differ across time [10] and when different kinds of vaccines are considered (i.e., mRNA vs. viral vectors vaccines [13]); however, we did not make such a distinction. 

Furthermore, possible limitations of the study should be considered. The adoption of an online survey might have discouraged the recruitment of elders, people with low education, or who had no easy access to the Internet. In addition, one might speculate that the higher the reluctance towards COVID-19 vaccination, the higher the chance that these more hesitant people would refuse to complete a questionnaire on the topic. Indeed, the COVID-19 vaccine acceptance was slightly higher in our sample than the overall observed adhesion in Italy in the period of testing [3]. Moreover, the snowball convenience sampling procedure adopted in this study possibly undermined the sample representativeness, relative to the prevalence of certain sociodemographic features in the overall Italian population. For instance, women, people with high education (i.e., a university degree or higher), and those living in Northern Italy were overrepresented [44]. This approach might weaken the generalizability of our findings, but it allowed a timely evaluation and guaranteeing social distancing. Furthermore, we adopted a cross-sectional design, which was inherent to the object of the investigation, but prevented the identification of causal effects among the variables considered [45]. However, the statistical approach used weighted the possible effects of each variable by the simultaneous effects of all the other variables, which is especially valuable when considering multidimensional and complex phenomena such as vaccine hesitancy. 

To conclude, our work extends the previous evidence on the psychological antecedents of COVID-19 vaccine hesitancy, clarifying how certain psychological dimensions might have affected the vaccine acceptance in Italy. Vaccine hesitancy is indeed a complex phenomenon; thus, the higher its comprehension the greater the possibility to tailor efficacious vaccine-related communications that drive people’s acceptance of vaccines. This may be crucial, not only to maintain an adequate immunity for COVID-19, but also to inform policymakers and public authorities in the case of possible future infectious outbreaks. 

## Figures and Tables

**Table 1 vaccines-10-01158-t001:** Means and standard deviations for all psychological variables examined in this study, separately for the entire sample (*n* = 1064) and for the two vaccine-related behavior groups (favorable, *n* = 928; hesitant, *n* = 136).

	Full Sample (*n* = 1064)	Hesitant (*n* = 136)	Favorable (*n* = 928)
COVID-19 Risk Perception	178.93 (61.8)	152.87 (66.39)	182.75 (60.19)
TIPI Agreeableness	5.40 (1.00)	5.36 (0.95)	5.40 (1.01)
TIPI Conscientiousness	5.54 (1.02)	5.40 (1.12)	5.57 (1.01)
H-LoC Internal	32.15 (4.44)	32.05 (4.79)	32.17 (4.39)
H-LoC External God	4.29 (2.28)	4.58 (2.46)	4.25 (2.25)
H-LoC External Others	5.63 (2.19)	5.79 (2.29)	5.61 (2.18)
CMQ	5.44 (1.83)	6.47 (1.58)	5.29 (1.82)
5-CPAV Confidence	5.54 (1.36)	3.44 (1.52)	5.84 (1.03)
5-CPAV Complacency	1.53 (0.84)	2.53 (1.21)	1.39 (0.66)
5-CPAV Constraints	1.52 (0.87)	1.87 (1.14)	1.47 (0.81)
5-CPAV Calculation	5.64 (1.43)	6.04 (0.91)	5.58 (1.48)
5-CPAV Collective Responsibility	6.56 (0.88)	5.33 (1.38)	6.74 (0.59)
PVQ-21 Conformity	−0.10 (0.82)	−0.41 (0.99)	−0.05 (0.78)
PVQ-21 Benevolence	0.68 (0.62)	0.66 (0.63)	0.68 (0.62)
PVQ-21 Self-Direction	0.36 (0.74)	0.63 (0.78)	0.32 (0.73)
PVQ-21 Security	0.19 (0.80)	0.12 (0.90)	0.20 (0.79)

Note. TIPI = Ten-Item Personality Inventory; H-LoC = Health Locus of Control scale; CMQ = Conspiracy Mentality Questionnaire; 5-CPAV = 5C Psychological Antecedents of Vaccination scale; PVQ = Portrait Value Questionnaire.

**Table 2 vaccines-10-01158-t002:** Results from the multiple hierarchical logistic regression on vaccine hesitancy (favorable, *n* = 928; hesitant, *n* = 136); significant *p*-values are reported in bold.

Variables	B	SE	Wald	*p*-Value	OR
Age	0.017	0.011	2.234	0.135	1.017
Sex	0.168	0.349	0.230	0.631	1.182
Civil Status, being in a relationship	−0.898	0.364	6.091	**0.014**	0.407
Education, university	−0.377	0.331	1.300	0.254	0.686
Education, Ph.D.	−0.516	0.433	1.421	0.233	0.597
SeS, medium income	0.905	0.371	5.947	**0.015**	2.472
SeS, high income	0.238	0.504	0.223	0.637	1.269
COVID-19 Contagion	−1.135	0.402	7.993	**0.005**	0.321
Being a healthcare professional	−0.112	0.171	0.434	0.510	0.894
TIPI Agreeableness	−0.180	0.154	1.360	0.243	0.836
TIPI Conscientiousness	0.190	0.143	1.758	0.185	1.210
H-LoC Internal	−0.017	0.034	0.238	0.626	0.983
H-LoC External God	0.007	0.065	0.013	0.909	1.007
H-LoC External Others	−0.041	0.067	0.375	0.540	0.960
CMQ	−0.136	0.095	2.027	0.154	0.873
COVID-19 Risk Perception	0.000	0.003	0.020	0.886	1.000
5C-PAV Confidence	3.867	0.434	79.361	**<0.001**	47.619
5C-PAV Complacency	−3.676	0.839	19.188	**<0.001**	0.025
5C-PAV Constraints	1.066	0.686	2.417	0.120	2.905
5C-PAV Calculation	−1.361	0.430	10.035	**0.002**	0.256
5C-PAV Collective Responsibility	2.915	0.568	26.343	**<0.001**	18.449
PVQ-21 Conformity	−0.301	0.201	2.259	0.133	0.740
PVQ-21 Benevolence	−0.008	0.246	0.001	0.974	0.992
PVQ-21 Self-Direction	−0.283	0.226	1.565	0.211	0.754
PVQ-21 Security	0.032	0.202	0.025	0.873	1.033
Constant	6.444	1.970	10.704	0.001	629.057

Note. Reference category is the favorable group (coded as 1). TIPI = Ten-Item Personality Inventory; H-LoC = Health Locus of Control scale; CMQ = Conspiracy Mentality Questionnaire; 5C-PAV = 5C Psychological Antecedents of Vaccination scale; PVQ = Portrait Value Questionnaire.

## Data Availability

The data presented in this study are available on request from the corresponding author. The data are not publicly available due to privacy reasons.

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
