# Peer review of "Behind the Scenes of COVID-19 Vaccine Hesitancy: Psychological Predictors in an Italian Community Sample"

_vaccines, 2022, doi:10.3390/vaccines10071158_

Round 1
Reviewer 1 Report
Dear Editor and Authors,
Thank you for asking me to review this manuscript titled “Behind the Scenes of COVID-19 Vaccine Hesitancy: Psychological Predictors in an Italian Community Sample.” by Dr. Tagini and her colleagues from Milan in Italy.
In this prospective cohort study of over 1000 Italian volunteers the authors investigated the psychological variables associated with vaccine hesitancy utilizing a variety of questionnaire techniques (web online and self reporting).
This is a well conducted study and I was particularly interested to see its results. The methodology is sound although I am unsure how the randomness of the sampling was ensured? Can the authors elaborate a bit on that!
The results are interesting and well-presented even though not really unexpected and the manuscript is well written in good quality language.
I think following some minor corrections this work does merit publication.
Kind regard,
Reviewer 2 Report
The authors have conducted an interesting study. I have some suggestions herein. There is a need to describe the methodology in various sections including study location, study design, sample size estimation, sampling techniques, development of data collection tool, validation and reliability of the study tool, components of the data collection form, outcomes measured, scoring criteria used in the study, and analysis. Please also provide the English translation of the study tool so it can benefit those who wish to use it in their own countries. There is no information on the reliability testing and validation of the study tool. The definitions of hesitant and favorable should be clear in the method section. The discussion section is very sound in this study. Please provide the study flow diagram.
Author Response
The authors have conducted an interesting study. I have some suggestions herein.
Reply: We are thankful for the Reviewer’s interesting and effort in revising our work. We have provided point-by-point responses to the Reviewer’s suggestions below.
Point 1: There is a need to describe the methodology in various sections including study location.
Reply: We are not sure what the Reviewer meant for study location. If the Reviewer referred to participant’s living place, in the Methods section we report “Most of the sample was living in Northern Italy (n = 999; 92.6%)” (line 97)”. In fact, this was listed among the possible limitations of our work (line 387): “Moreover, the snowball convenience sampling procedure adopted in this study possibly undermined the sample representativeness, relative to the prevalence of certain sociodemographic features in the overall Italian population. For instance, women, people with high education (i.e., a university degree or higher) and living in Northern Italy were overrepresented [41]”. However, if the Reviewer was referring to a different point, we kindly ask to further specify the request.
Point 2: study design.
Reply: As reported in Methods section (line 104): “An online, cross-sectional survey was conducted between April and May 2021, approximately three months after the start of the vaccinal campaign in Italy.”. Also, the possible limitations of this study approach are discussed (line 393): “Also, we adopted a cross-sectional design, which is inherent to the object of the investigation but prevents the identification of causal effects among the variables considered [42]. However, the statistical approach used weights the possible effect of each variable by the simultaneous effect of all the other variables, which is especially valuable when considering multidimensional and complex phenomena such as vaccine hesitancy.”
Point 3: sample size estimation.
Reply: We thank the Reviewer for noting this. We have now reported sample size estimation at the beginning of the Methods (Participants) section, as submitted to our Ethical Committee. Please see line 89: “A priori sample size estimation performed with G*Power software [20] showed that 1045 participants would be sufficient in order to achieve the 95% of statistical power, considering a logistic regression analysis with predictors moderately correlated (r2 = 0.25), an average odds ratio of 1.30 and a given alpha level of 0.05.”
Point 4: sampling techniques
Reply: Concerning the sampling procedure, we report in the Methods section that “A snowball convenience sampling procedure was adopted” (line 107). We acknowledge this approach may undermine the sample representativeness; indeed, when discussing on the possible study limitations we specify (line 387): “Moreover, the snowball convenience sampling procedure adopted in this study possibly undermined the sample representativeness, relative to the prevalence of certain sociodemographic features in the overall Italian population. For instance, women, people with high education (i.e., a university degree or higher) and living in Northern Italy were overrepresented [41]. This approach might weaken the generalizability of our findings, but it allowed a timely evaluation, guaranteeing social distancing”. If the mentioned information may not be satisfying, we may kindly ask to further detail this request to accomplish to this point more exhaustively.
Point 5: development of data collection tool
Reply: We thank the Reviewer for pointing this out. The measures included in the study were part of a general survey on COVID-19 vaccine acceptance in Italy. We have now specified this point (line 81): “This study aims to extend the previous literature by outlining more comprehensively the psychological profile associated with COVID-19 vaccine hesitancy in Italy, evaluating in a sample of Italian citizens those psychological variables previously, but independently, reported to be significant predictors of COVID-19 vaccine hesitancy. To this purpose, several self-report psychological measures were collected as part of a web survey on COVID-19 vaccine acceptance released in Italy between April and May 2021.”
Also, in the Methods section we have detailed the procedure ad follow (line 119): “An online form was developed according to the previous literature on vaccine hesitancy, in general, and relative to COVID-19 vaccine [8,21–24]: those predictors identified as most influential were included and assessed through ad hoc questions or standardized measures. Eventually, several items were included in the form, including sociodemographic, health-related, behavioral, and psychological information (see Supplementary Material Table S1 for details on the composition of the overall survey). Information and measures to be included in the present study were selected in line with the study aim and according to the available literature on the psychological predictors of COVID-19 vaccine hesitancy [10,11,13,14,16–19]; they are detailed below. Furthermore, essential sociodemographic information was included in the analysis: specifically, age, sex, place of living, civil status, level of education, profession, socio-economic status, and certified COVID-19 contagion.”
As required by the Reviewer (Point 9) the English translation of the web survey was reported for clarity’s sake and to be possibly used in future investigations.
Point 6: validation and reliability of the study tool
Reply: We thank the Reviewer for pointing this out, further details have been provided. Concerning the Risk Perception measure, which was developed by our group in a previous study, we have now specified: “Risk perception for COVID-19 was investigated with a 5-items questionnaire previously developed by our research group, which was proved of good internal validity [20]”.
Also, we now report that (line 139): “Italian validated translations of standardized instruments, with verified psychometric properties, were adopted to measure additional psychological constructs. Those instruments that were not available in Italian were translated using a forward and backward translation procedure [23]; the Italian translations adopted are reported in the Supplementary Material. Also, Confirmatory Factor Analyses were performed to probe the internal validity of the Italian version of the translated measures (see the Supplementary Material). For all measures, Cronbach’s alpha or inter-item correlation coefficients are provided as indices of internal reliability.”
More specifically, concerning the 5C-PAV scale: “a Confirmatory Factor Analysis (CFA) evidenced that the Italian version of the 5C-PAV had good internal validity (see Supplementary Material)”. Also, concerning the CMQ scale: “a Confirmatory Factor Analysis (CFA) evidenced that the Italian version of the CMQ had good internal validity (see Supplementary Material)”.
Point 7: components of the data collection form
Reply: We apologize, but we are not sure which is the Reviewer’s request; thus, we kindly ask to detail further this point. Anyway, all the questions and measures included in the form have been provided and descripted in the Methods (Measures section).
Point 8: outcomes measured, scoring criteria used in the study, and analysis.
Reply: Details on the outcomes (including subscales computed and scoring criteria) and the analysis performed are provided in the Methods section; anyway, we have now specified that “Standardized instruments were scored according to guidelines”. If the Reviewer does not consider the information provided exhaustive, we kindly ask to detail his/her request about the information we should further provide, to better accomplish to this point.
Point 9: Please also provide the English translation of the study tool so it can benefit those who wish to use it in their own countries.
Reply: The English translation of the web survey on COVID-19 vaccine acceptance was reported for clarity’s sake and to be possibly used in future investigations in the Supplementary Material Table S1.
Point 10: There is no information on the reliability testing and validation of the study tool.
Reply: Please, see our reply to point 6.
Point 11: The definitions of hesitant and favorable should be clear in the method section.
Reply: As reported in the Methods section (line 210): “We identified two groups, according to participants’ willingness to be vaccinated. Participants who have already got the shot or who were willing to be vaccinated represented the Favorable group; those who were doubtful, who were certainly not going to be vaccinated in the future, or who got (or will get) the shot only because it was mandatory for working represented the Hesitant group”. If the Reviewer may not consider this description exhaustive, we kindly ask to provide further suggestions.
Point 12: Please provide the study flow diagram.
Reply: We were not sure about what the Reviewer was referring to: was the Reviewer meaning the study flow diagram concerning the participants selection/exclusion process? We have now provided details of the number of participants excluded with relative reasons in the Methods (Participants section): “A total of 1122 online records were obtained; 1079 participants were included in the final sample (mean age: 46.91 ± 15.59 years, age range: 18-89; 788 females, 73.2%) after removing a few duplicated cases (n = 4), those who did not agree with the privacy regulation (n = 33), and those who did not live in Italy (n = 6)”, but if this was not what was required, we kindly ask further clarifications.
